# lo-fi: distributed fine-tuning without communication

**Mitchell Wortsman**[*]                                                    *mitchnw@cs.washington.edu*
*University of Washington*

**Suchin Gururangan**                                                    *sg01@cs.washington.edu*
*University of Washington*

**Shen Li**                                                                      *shenli@meta.com*
*Meta AI Research, FAIR Team*

**Ali Farhadi**                                                               *ali@cs.washington.edu*
*University of Washington*

**Ludwig Schmidt**                                                      *schmidt@cs.washington.edu*
*University of Washington*

**Michael Rabbat**                                                       *mikerabbat@meta.com*
*Meta AI Research, FAIR Team*

**Ari S. Morcos**                                                          *arimorcos@meta.com*
*Meta AI Research, FAIR Team*

**Reviewed on OpenReview:** *https://openreview.net/forum?id=1U0aPkBVz0&referrer=%5BTMLR%5D*

## Abstract

When fine-tuning large neural networks, it is common to use multiple nodes and to communicate gradients at each optimization step. By contrast, we investigate completely local fine-tuning, which we refer to as lo-fi. During lo-fi, each node fine-tunes independently without any communication. Then, the weights are averaged across nodes at the conclusion of fine-tuning. When fine-tuning DeiT-base and DeiT-large on ImageNet, this procedure matches accuracy in-distribution and improves accuracy under distribution shift compared to the baseline, which observes the same amount of data but communicates gradients at each step. We also observe that lo-fi matches the baseline's performance when fine-tuning OPT language models (up to 1.3B parameters) on Common Crawl. By removing the communication requirement, lo-fi reduces resource barriers for fine-tuning large models and enables fine-tuning in settings with prohibitive communication cost.

## 1 Introduction

Many of the best performing machine learning models today come from a two step procedure: First, *pre-train* on a large, heterogeneous dataset to learn a good representation. Next, *fine-tune* to adapt the model to a task of interest (Girshick et al., 2014; Yosinski et al., 2014; Kornblith et al., 2019; Kolesnikov et al., 2020). This paper operates within the second step of this procedure—fine-tuning—which is increasingly important with drastic improvements in pre-trained models, e.g., CLIP (Radford et al., 2021), GPT-3 (Brown et al., 2020), OPT (Zhang et al., 2022), and PaLM (Chowdhery et al., 2022). Indeed, recent advances such as Minerva (Lewkowycz et al., 2022) or InstructGPT (Ouyang et al., 2022) have come from fine-tuning rather than training from scratch.

---

[*]Work done while MW and SG were at FAIR.

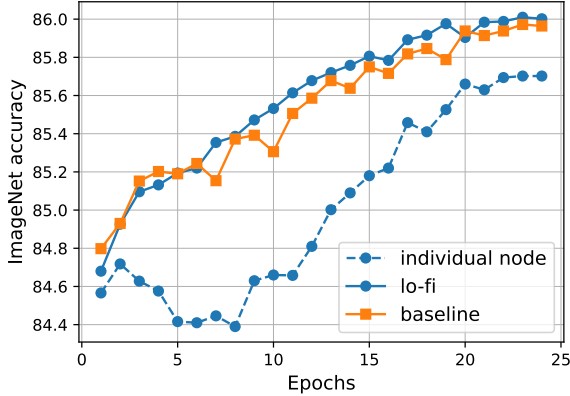

Figure 1: In standard multi-node distributed data-parallel fine-tuning, there is synchronization between nodes at each step of fine-tuning. With lo-fi (local fine-tuning), there is no communication between nodes throughout fine-tuning. As a result, each node $k$ independently produces their own model $\theta^k$. Then, lo-fi averages these models once for the final solution $\theta_{\text{lo-fi}} = \frac{1}{n}\sum_{k=1}^{n}\theta^k$. In this four-node fine-tuning run, we show (i) the average accuracy of the individual models $\theta^k$, (ii) the accuracy of $\theta_{\text{lo-fi}}$ at the end of each fine-tuning epoch, and (iii) the accuracy of the baseline which communicates among nodes every step. In particular, we fine-tune the ImageNet-21k pre-trained DeiT-base model from DeiT-III (Touvron et al., 2022) on ImageNet (Deng et al., 2009) using their code, which uses four nodes.

|  | IN | IN-V2 | IN-R | Sketch | IN-A |
|---|---|---|---|---|---|
| baseline (DeiT-b) | 85.96 | 76.65 | 62.66 | 46.86 | 57.15 |
| lo-fi (DeiT-b) | 86.00 | 76.84 | 63.25 | 48.37 | 58.43 |
| baseline (DeiT-l) | 87.12 | 78.18 | 69.87 | 54.41 | 68.97 |
| lo-fi (DeiT-l) | 87.10 | 78.25 | 70.14 | 54.95 | 69.53 |

Table 1: Comparing lo-fi (no communication during fine-tuning) to the baseline which communicates at each step when fine-tuning the ImageNet-21k pre-trained DeiT-base and DeiT-large model from DeiT-III (Touvron et al., 2022) on ImageNet (Deng et al., 2009). Both lo-fi and the baseline use the same number of iterations, which have been tuned for the baseline. Underlined numbers indicate significantly better accuracy according to McNemar's test with significance level 0.05. Lo-fi matches performance on ImageNet (IN), but can outperform the baseline on some distribution shifts. The shifts we consider are IN-V2 (Recht et al., 2019), IN-R (Hendrycks et al., 2021a), Sketch (Wang et al., 2019), and IN-A (Hendrycks et al., 2021b).

Most work developing learning methods still operates in the paradigm of training from scratch. Accordingly, both use similar algorithmic techniques despite important differences in the pre-training and fine-tuning regimes. In particular, one notable difference between pre-training and fine-tuning is that fine-tuned models appear to lie in a single low-error region (Neyshabur et al., 2020). Indeed, linearly interpolating the weights of fine-tuned models can have similar advantages as ensembling their predictions but without the added cost during inference (Wortsman et al., 2022). By contrast, linearly interpolating the weights of two models trained from scratch will encounter a high error barrier (Frankle et al., 2020; Garipov et al., 2018).

Recently, the model soups approach (Wortsman et al., 2022) leveraged this similarity between ensembling outputs and averaging weights. Given a hyperparameter sweep over fine-tuned models, they average the weights of multiple models instead of the conventional procedure of selecting one model and discarding the remainder. However, the model soups approach does not modify the fine-tuning procedure itself.

In this paper, we leverage the observation that fine-tuned models appear to lie in a single low error region to remove communication between nodes during distributed fine-tuning. In standard data-parallel multi-node fine-tuning, gradients between nodes are communicated at each step. This synchronization of updates keeps the models at each node identical to each other during fine-tuning. However, in certain settings communication costs during fine-tuning may be prohibitive, and we therefore ask whether they are necessary at all. With our method of local fine-tuning, which we refer to as *lo-fi*, we remove all communication between nodes during fine-tuning. The models on each node therefore drift apart throughout fine-tuning. Then, to arrive at the final solution at the end, we average the weights of the models produced by each node.

|  | IN | IN-V2 | IN-R | Sketch | IN-A | epochs | stoch. depth | no extra cost | no comms |
|---|---|---|---|---|---|---|---|---|---|
| DeiT-base |  |  |  |  |  |  |  |  |  |
| (Touvron et al., 2022) | 85.72 | 76.53 | 61.83 | 47.44 | 57.29 | 50 | 0.15 | ✓ |  |
| baseline | 85.96 | 76.65 | 62.66 | 46.86 | 57.15 | 24 | 0.15 | ✓ |  |
| individual node | 85.66 | 76.22 | 62.09 | 46.75 | 56.00 | 6 | 0.10 | ✓ | ✓ |
| lo-fi | 86.00 | 76.84 | **63.25** | **48.37** | **58.43** | 24 | 0.10 | ✓ | ✓ |
| lo-fi ensemble | **86.08** | **76.91** | 63.05 | 47.67 | 57.80 | 24 | 0.10 |  | ✓ |
| DeiT-large |  |  |  |  |  |  |  |  |  |
| (Touvron et al., 2022) | 86.97 | 78.47 | 69.70 | 54.35 | 68.57 | 50 | 0.40 | ✓ |  |
| baseline | 87.12 | 78.18 | 69.87 | 54.41 | 68.97 | 12 | 0.30 | ✓ |  |
| individual node | 86.76 | 78.00 | 69.41 | 54.57 | 67.59 | 3 | 0.25 | ✓ | ✓ |
| lo-fi | 87.10 | 78.25 | **70.14** | **54.95** | **69.53** | 12 | 0.25 | ✓ | ✓ |
| lo-fi ensemble | **87.14** | **78.35** | 70.00 | 54.62 | 69.20 | 12 | 0.25 |  | ✓ |

Table 2: Expanding the comparison between lo-fi and the baseline (Table 1) when fine-tuning the ImageNet-21k pre-trained models from DeiT-III (Touvron et al., 2022) on ImageNet (Deng et al., 2009). In this four node fine-tuning run, lo-fi removes communication between nodes so that each node produces an independent model. The weights of the models are then averaged at the end to produce the final solution. In this table we bold the highest number and evaluate the following models: i) paper, the fine-tuned models from the DeiT-III paper (Touvron et al., 2022), ii) baseline, which is our improved fine-tuning baseline after hyperparameter turning which requires less epochs of training but achieves slightly higher accuracy than reported in the DeiT-III paper (Touvron et al., 2022), iii) individual node, which is one of the individual node models that is produced by lo-fi, iv) lo-fi, which fine-tunes individual models on each node then averages their weights once at the end, and v) lo-fi ensemble which averages the outputs of the models produced by each node during lo-fi, and therefore requires more cost during inference. In addition to evaluating on ImageNet (IN), the task used for fine-tuning, we also evaluate on the distribution shifts ImageNet-V2 (IN-V2, (Recht et al., 2019)), ImageNet-R (IN-R, (Hendrycks et al., 2021a)), ImageNet-Sketch (Wang et al., 2019), and ImageNet-A (IN-A (Hendrycks et al., 2021b)). While more information is provided in Section 3.1, this table also displays some hyperparameter changes we made from the default DeiT-III fine-tuning script. Unlike (Touvron et al., 2022), we fine-tune with LP-FT (Kumar et al., 2022), and observe it is better for the baseline to use fewer fine-tuning epochs. Lo-fi observes the same amount of data as the tuned baseline and uses the same hyperparameters with the exception of slightly decreased regularization by lowering stoch. depth (Huang et al., 2016) drop probability by 0.05 (making the same change to the baseline decreased accuracy). Additional columns track whether the model incurs no additional cost during inference compared to a single model (denoted no extra cost), and also if there is no communication between nodes during fine-tuning (denoted no comms). Overall, lo-fi matches or outperforms the baseline without communication between nodes during fine-tuning.

We note that these techniques are a natural extension of previous work: lo-fi is just a *model soup* (Wortsman et al., 2022) formed by splitting up a large fine-tuning job into multiple smaller jobs, each isolated to a node. Analogously, lo-fi is embarrassingly parallel training from *branch-train-merge* (Li et al., 2022) applied in the setting where no domain specialization info is provided and so each expert is trained on IID data. However, we believe that the application of these techniques in this setting is of practical interest, especially if models continue to grow.

In computer vision we use the DeiT-III codebase (Touvron et al., 2022) to fine-tune the ImageNet-21k pre-trained DeiT-base and DeiT-large models, which are four-node fine-tuning jobs by default. We observe (Figure 1, Table 1) that lo-fi matches the accuracy of DeiT-base and DeiT-large on ImageNet, the task used for fine-tuning, while outperforming the baseline on some distribution shifts. These improvements come after hyperparameter tuning the baseline to slightly exceed that in the DeiT-III paper while requiring fewer fine-tuning epochs. Moreover, lo-fi and the baseline observe the same amount of data. While overall similar results are observed when fine-tuning CLIP ViT-L (Radford et al., 2021) on ImageNet or tasks from WILDS (Koh et al., 2021), lo-fi often requires more iterations in this setting. Finally, we test lo-fi beyond computer vision by fine-tuning OPT-125M and OPT-1.3B (Zhang et al., 2022) on Common Crawl, observing that lo-fi can match the baseline which communicates between nodes.

Overall, our work is a test of whether communication between nodes is required during fine-tuning. However, we also wanted to understand the advantages of removing this communication. Therefore, we benchmark the wall-clock overhead of communication on an AWS cluster with EFA. We use the models from the DeiT-III repository (Touvron et al., 2022) in the context of image classification. In this setting and on the system used for this study, the advantages are overall less substantial than we initially expected, especially for large batch sizes. Notably, we observe that the trick of overlapping the communication and computation in the backwards pass (Li et al., 2020a), which is the default in PyTorch (Paszke et al., 2019) as of v1.5, reduces the overhead of using multiple nodes from roughly 50% slow-down to under 10% for the large DeiT model. Finally, we discuss how lo-fi can help with faster job scheduling and addresses the straggler and jitter problem in distributed training, where different nodes might experience random slowdowns.

## 2  Methods

This section details the methods used in our experiments. We begin with the baseline of standard data-parallel training, and next outline our straightforward modification which i) removes communication between nodes then ii) averages the final models produced by each node.

Consider a neural network $f(x, \theta)$ where $x$ is the input data and $\theta \in \mathbb{R}^d$ are the network parameters. Since we are fine-tuning, $\theta$ is initialized as the weights of a pre-trained model. Moreover, as is standard in neural network training, the input data $x$ is a batch rather than a single data point. Finally, let $n$ denote the number of devices, $b$ denote total batch size, and $\ell(\hat{y}, y)$ denote loss for the vector of predicted labels $\hat{y} = f(x, \theta)$ and a vector of ground-truth labels $y$.

**With communication.** The most straightforward and common approach for training with $n$ devices is data-parallel. In this setting, each device has their own copy of the parameters $\theta$. During fine-tuning, each batch $x$ of size $b$ is split into $n$ disjoint sub-batches of size $b/n$. Each device $i$ loads the sub-batch $(x_i, y_i)$ and computes gradients $g_i = \nabla_\theta \ell(f(x_i, \theta), y_i)$. Then the gradients are synchronized across nodes with each node computing an averaged gradient $\bar{g} = \frac{1}{n} \sum_{i=1}^{n} g_i$. After synchronizing gradients, each device uses $\bar{g}$ to update $\theta$. Since every device updates $\theta$ using an identical gradient $\bar{g}$, the parameters $\theta$ remain identical across devices.

**lo-fi.** With local-finetuning (lo-fi), we partition the $n$ devices into $K$ disjoint groups. In the majority of our experiments, each group is a single node containing 8 GPU devices. During fine-tuning we allow communication within each group, but not across groups. Each group $k$ begins with parameters $\theta^k$ which are initially identical across devices, but drift apart throughout fine-tuning. Then, at the end of fine-tuning there is a single communication and the parameters from each group are averaged to produce a final solution $\theta = \frac{1}{K} \sum_{k=1}^{K} \theta^k$.

There are two possible implementations for lo-fi which we refer to as implementation A and B. Implementation A proceeds as before—each device $i$ loads the sub-batch $(x_i, y_i)$ and computes gradients $g_i$. There is then gradient synchronization only among devices belonging to the same group while devices from different groups apply different gradients. Data partitioning is accomplished without communication by coordinating random seeds, so long as each device knows its rank and the total number of devices. Our experiments primarily use Implementation A.

In Implementation B, each group is a completely independent run—no knowledge of total number of devices is required. Accordingly, within each group the global batch size is scaled by $1/K$ so that the per-device batch size is matched. Our image classification results use Implementation A while our language modelling results use Implementation B.

Our motivation for having one group per node and still allowing communication among the devices on the node is that communication within a node is faster than communication across nodes.

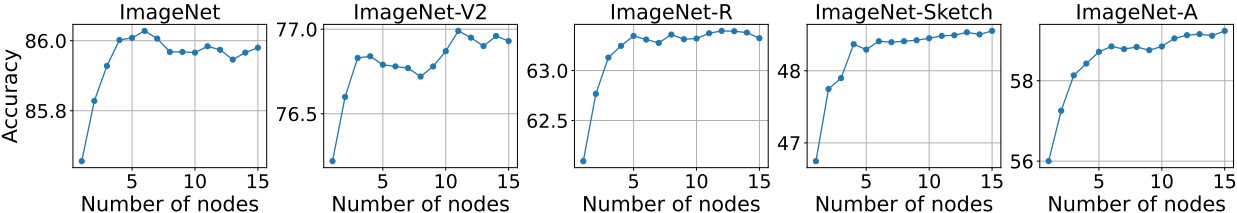

Figure 2: We test whether the performance of lo-fi continues to improve when adding more nodes. On the contrary, this experiment suggests diminishing or even negative returns after 4 nodes. This experiment is for fine-tuning DeiT-base as in Table 1. Recall that when using four nodes, lo-fi and the baseline observe the same number of images, but lo-fi does not require communication between nodes. When moving beyond 4 nodes as we do in this experiment, lo-fi observes more images then the baseline.

## 3 Experiments

This section presents our experiments which test whether communication is required during fine-tuning. First we use the DeiT-III codebase (Touvron et al., 2022) to fine-tune their pre-trained ImageNet-21k models on ImageNet, where we observe that lo-fi matches the baseline but without communication between nodes (Section 3.1). Next, we fine-tune CLIP (Radford et al., 2021) on ImageNet, WILDS-FMoW (Koh et al., 2021; Christie et al., 2018) and WILDS-iWildCam (Beery et al., 2021) (Section 3.2). Finally, we show preliminary experiments applying lo-fi outside of computer vision (Section 3.3) and benchmark the associated speed-ups by removing communication (Section 3.4).

### 3.1 Fine-tuning DeiT-III on ImageNet

The aim of these experiments is to test whether communication between nodes is required when fine-tuning high accuracy models for image classification. To test this we begin by fine-tuning the DeiT-base and DeiT-large models from the DeiT-III paper (Touvron et al., 2022) using their code. In particular, we fine-tune their ImageNet-21k models on ImageNet-1k (Deng et al., 2009) with and without lo-fi.

We chose the models from DeiT-III for a few reasons: (i) DeiT-III is representative of state-of-the-art settings as it uses many advanced techniques such as stochastic depth (Huang et al., 2016), CutMix (Yun et al., 2019), and the LAMB optimizer (You et al., 2019). (ii) DeiT-III provides hyperparameter configurations which they used in their fine-tuning experiments. (iii) DeiT-III uses 4 nodes with 8 GPUs each when fine-tuning their pre-trained ImageNet-21k models on ImageNet. This provides an opportunity to test lo-fi in an equivalent setting where there is normally communication between nodes.

**Main results.** Our overall finding is that communication between nodes is not necessary in this setting—lo-fi matches the accuracy of the baseline while observing the same amount of data. These results are presented in Figure 1 and Tables 1 and 2. In these experiments, lo-fi uses 4 groups—each group corresponds to one node.

Figure 1 illustrates accuracy throughout training when fine-tuning DeiT-base with and without lo-fi. We also report the average accuracy of the models produced by the individual nodes. To make this plot we display the accuracy of the averaged lo-fi model at the end of each epoch, though usually we would only average the models once at the end. A question emerges when looking at this plot: why does the accuracy of the individual node first dip before coming back up? The answer is due to the interaction of learning rate and batch size, which we discuss further in Appendix B.

Table 1 evaluates the final models from Figure 1 on ImageNet as well as under distribution shift (on ImageNet-V2 (Recht et al., 2019), ImageNet-R (Hendrycks et al., 2021a), ImageNet Sketch (Wang et al., 2019), and ImageNet-A (Hendrycks et al., 2021b)). In addition, Table 1 repeats the experiment from Figure 1 with the DeiT-large model. We underline any result that is significantly better (using McNemar's test with significance

0.05). Overall we observe that lo-fi matches the accuracy of the baseline which uses communication, and outperforms the baseline under distribution shift.

Table 2 supplements Table 1 with additional details. In particular, we consider the accuracy of the model produced by an individual node during lo-fi, before the averaging. We also evaluate the output-space ensemble of the models produced by each node during lo-fi, which is more expensive during inference as a pass through each model is required. Finally, we display the accuracy of the models fine-tuned in the DeiT-III paper (Touvron et al., 2022). We improved our own baseline over that in the paper with the following hyperparemter changes: (i) Instead of removing the classification layer of the pre-trained model, we implement a version of LP-FT (Kumar et al., 2022) to fine-tune—we preserved the ImageNet-21k classifier then use a class mapping from ImageNet-21k to ImageNet classes. (ii) We remove the grayscale, solarization, and Gaussian blur augmentations, since we found this improves accuracy. This aligns with previous research where fine-tuning requires less augmentation (Wortsman et al., 2022). (iii) We fine-tuned for fewer epochs, which also required a switch to a cosine scheduler that updates every iteration instead of every epoch so the schedule could complete. We also considered different values for the learning rate and stochastic depth, but found the default values to be best (Touvron et al., 2022). This is with the exception of DeiT-large for which we found stochastic depth 0.3 to be better for the baseline, which is what we used.

Lo-fi was run using identical hyperparameters except we decreased the stochastic depth drop rate by 0.05 for both DeiT-base and DeiT-large since each node is effectively fine-tuning on less data and may therefore require less regularization. The most substantive change from the DeiT-III code was to use LP-FT (Kumar et al., 2022), which we accomplished by preserving the classification layer from the pre-trained model and using a mapping from ImageNet-21k to ImageNet[1]. While this change results in a minor improvement for the baseline, we found it was necessary for achieving matching performance with lo-fi. Overall, despite the extensive hyperparameter tuning we performed for the baseline, lo-fi was still able to match or exceed the accuracy.

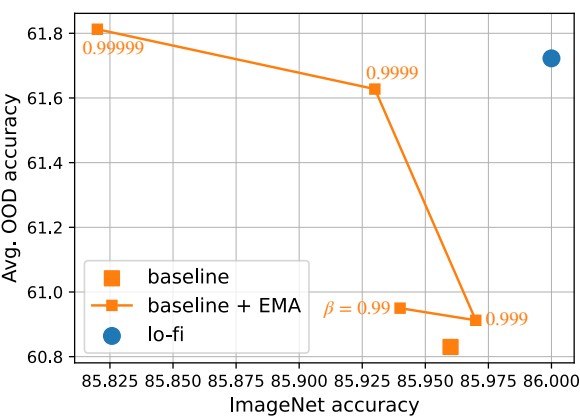

Figure 3: For the experiment in Table 1, lo-fi outperforms the baseline under distribution shift. We wanted to test whether this OOD performance ($y$-axis) could be improved by applying weight averaging techniques to the baseline. We observe that the answer is yes with EMA (Szegedy et al., 2016), although this can come at slight cost in-distribution accuracy ($x$-axis). In this plot we try 4 different values of EMA decay $\beta$. Applying EMA to lo-fi had minimal benefit, as did applying WiSE-FT (Wortsman et al., 2021) to the baseline. The ImageNet-21k$\rightarrow$ImageNet transfer setting is not characteristic of those studied in the WiSE-FT paper.

**Ablations.** We ran three ablation studies to better understand the performance of lo-fi in this setting. First, we wanted to test whether adding more nodes was helpful. In the initial 4 node experiment with lo-fi, we matched the baseline in terms of total amount of data observed, allowing a fair compute-matched comparison. However, there are practical settings such as privacy-preserving ML in which the benefits of reduced communication may outweigh the importance of matched compute. In Figure 2 we observed that adding more nodes did not improve in-distribution accuracy. Interestingly, however, adding additional nodes marginally improved out-of-distribution performance, most notably on ImageNet-Sketch and ImageNet-A.

Next, we wanted to understand if four groups, one per node, was optimal in this setting. What happens if we instead use 8 groups—2 per node, or 2 groups—each group consisting of 2 nodes? In this experiments the amount of data observed remains constant; all that changes is the amount of communication. As presented

---

[1]The only class in ImageNet but not ImageNet-21k is *teddy bear*—we initialize this row with *bear* instead.

| Groups | 2 | 4 | 8 | 16 |
|---|---|---|---|---|
| ImageNet Accuracy | 85.95 | 86.00 | 85.85 | 85.73 |

Table 3: For our four-node fine-tuning jobs, we usually partition the 32 GPUs into 4 communication groups, one per-node. This table shows the effect of partitioning the GPUs into groups of different sizes, finding slightly worse performance when the number of groups is large.

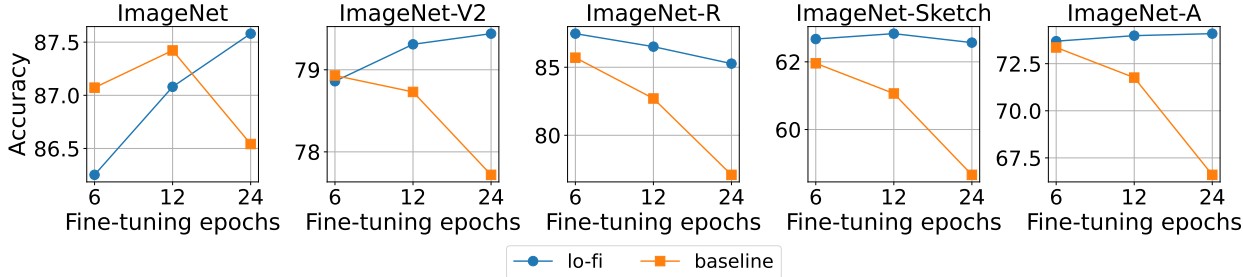

Figure 4: We fine-tune CLIP ViT-L (Radford et al., 2021; Dosovitskiy et al., 2021) on ImageNet. In contrast to the DeiT fine-tuning experiments, the models were not pre-trained with stochastic depth and we found better accuracy when fine-tuning without stochastic depth. Instead, we fine-tune for 6, 12, and 24 epochs. lo-fi shows good performance under distribution shift, but on ImageNet requires more epochs to exceed the baseline accuracy unlike in the DeiT experiments.

in Table 3, accuracy drops slightly when using a larger number of groups[2]. This result demonstrates that the best configuration is one group per node.

Finally, we found it interesting the lo-fi outperformed the baseline under distribution shift. Accordingly, we wanted to test whether we could recover these out-of-distribution (OOD) improvements by applying other weight averaging techniques to the baseline. We observe in Figure 3 that the answer is yes, although at slight cost to in-distribution performance for the methods we tried. The best performing technique we tried was a debiased exponential moving average (EMA) (Szegedy et al., 2016; Kingma & Ba, 2014), for which we tried decay values 0.99, 0.999, 0.9999, and 0.99999. We also tried applying EMA and WiSE-FT (Wortsman et al., 2021) to lo-fi, but did not observe out-of-distribution improvements [3].

## 3.2 Fine-tuning CLIP ViT-L on ImageNet and WILDS

In the previous section we observed that lo-fi matches the baseline for DeiT-III on ImageNet, but how does lo-fi perform for models pre-trained on larger datasets? In this section, we further test lo-fi for the CLIP ViT-L (Radford et al., 2021; Dosovitskiy et al., 2021) when fine-tuning on ImageNet (Figure 4) as well as two datasets from WILDS (Koh et al., 2021) (Figure 5).

Unlike the DeiT models, CLIP was not pre-trained with stochastic depth and we find better accuracy when we fine-tune without stochastic depth. This is unlike the DeiT-III models, which we found performed best when we used some stochastic depth. Indeed, this allowed us to use slightly less regularization for lo-fi then we did for the baseline by decreasing stochastic depth drop rate by 0.05. As this is no longer the case, we instead show experiments when fine-tuning for different numbers of epochs. Other than this omission of stochastic depth and varying the training epochs, the hyperparameter configuration is identical to that discussed in the previous section and follows the ImageNet-21k→ImageNet fine-tuning set-up from DeiT-III (Touvron et al., 2022).

---

[2]When using 2, 8, and 16 groups we changed the stochastic depth drop rate by 0.05, -0.05, and -0.10, respectively, from the four group setting.

[3]The intuition from WiSE-FT (Wortsman et al., 2021) is that of combining a generalist and specialist. Our intuition for why WiSE-FT does not show substantial improvements in the ImageNet-21k→ImageNet transfer setting is because both models are ImageNet specialists.

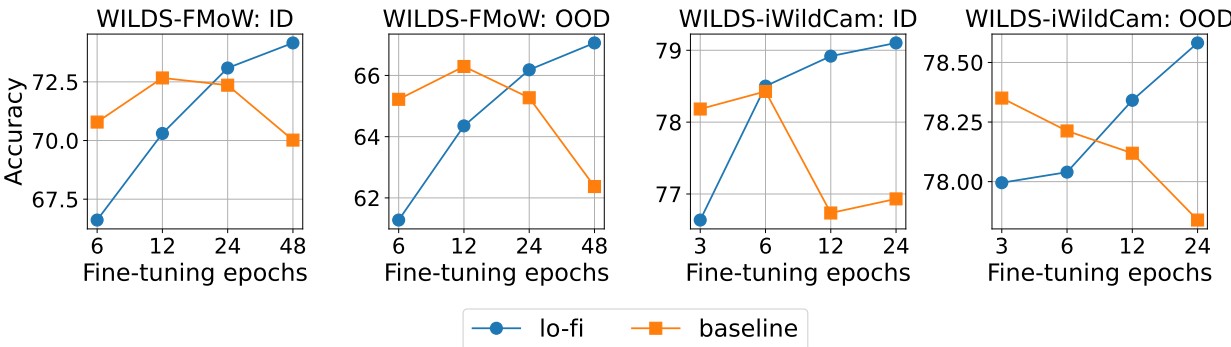

Figure 5: We repeat the CLIP ViT-L fine-tuning experiment from Figure 4 one two other image classification tasks: WILDS-FMoW (Koh et al., 2021; Christie et al., 2018), a satelite recognition task with a geographic and temporal distribution shift and WILDS-iWildCam (Koh et al., 2021; Beery et al., 2021), a camera trap dataset with a geographic distribution shift. Overall, we find similar results as in Figure 4.

Results when fine-tuning CLIP ViT-L on ImageNet are presented in Figure 4. For this experiment, we initialize the classification head of the zero-shot model using the zero-shot classifier output by the CLIP text tower (as in Wortsman et al. (2021)). We observe that more fine-tuning epochs are required for lo-fi to outperform the baseline on ImageNet. Under distribution shift, lo-fi roughly matches or exceeds the baseline for each of the fine-tuning epochs we tried. While this result indicates that lo-fi is a promising alternative to the baseline in this setting, a key limitation is that additional fine-tuning epochs were required to enable this improvement. The accuracy improvements beyond the best baseline model are consistent with the results reported in model soups (Wortsman et al., 2022).

We also test CLIP ViT-L on two further datasets, WILDS-FMoW (Koh et al., 2021; Christie et al., 2018), a satellite image recognition dataset with a temporal distribution shift and WILDS-iWildCam (Koh et al., 2021; Beery et al., 2021), a classification dataset with camera traps in the wild with a geographic distribution shift. Our motivation is to test lo-fi on natural images beyond the ImageNet universe. The results are presented in Figure 5, observing very similar results to the aforementioned experiment of fine-tuning CLIP ViT-L on ImageNet. However, there is an important difference in the experimental set-up. For these experiments, we first tried using the zero-shot initialization for the last layer of the model, as we did with ImageNet. However, this resulted in worse accuracy for lo-fi. Accordingly, these experiments are completed using the LP-FT method of fine-tuning (Kumar et al., 2022). First, we train a linear probe using one node. This linear probe is then used as the initialization when end-to-end fine-tuning individually on each node. We also apply this change to the baseline, but the benefit is much less substantial for the baseline than for lo-fi. Finally, for this experiment we used learning rate 7e-4 which we found resulted in higher accuracy for lo-fi and the baseline.

### 3.3 Language model fine-tuning

We also test lo-fi outside of image classification by fine-tuning OPT-125M and OPT-1.3B (Zhang et al., 2022).

**Experimental Setup** We report i)individual node, which is the average performance of the models produced by each node when using lo-fi, ii) lo-fi, which averages models produced by each node, and iii) baseline, which uses communication between nodes. For the 125M parameter model, we set the learning rate to 6e-5, with 1024-length sequence blocks, and 500K tokens per batch. For the 1.3B parameter model, we set the learning rate to 1e-5, with 512-length sequence blocks, and 1M tokens per batch. We use fp16 mixed precision (Micikevicius et al., 2017) for all experiments. We fine-tune the 125M parameter model with 4 nodes, and we fine-tune the 1.3B parameter model with 8 nodes. When using lo-fi there is no communication between nodes, so the experiments produce 4 and 8 models, respectively. Each node consists of 8 Volta 32GB GPUs connected with 400GBps interconnect.

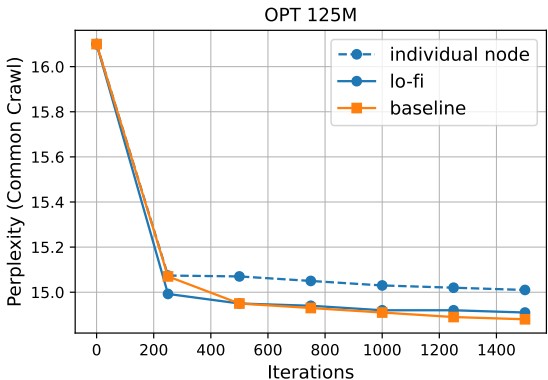 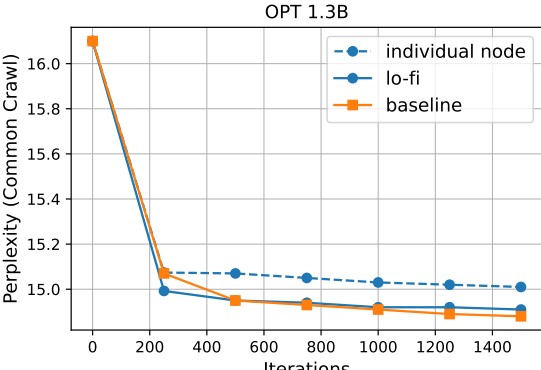

Figure 6: Fine-tuning a language model (left: OPT-125M, right: OPT-1.3B) on Common Crawl with lo-fi closely approaches the performance of the baseline of multi-node fine-tuning with communication. Here, we train four lo-fi workers independently, one per node. The baseline consists of standard data-parallel fine-tuning using four nodes, where there is communication between nodes at every iteration. The $x$-axis shows iterations, which does not take into account that lo-fi may be faster.

**Results** We fine-tune on the Pile's Common Crawl subset (Gao et al., 2021) using the Huggingface Transformers library (Wolf et al., 2020). Results are presented in Figure 6. We observe that for both model scales, when comparing by step count, lo-fi roughly matches the performance of the baseline, providing large performance improvements over the individual node setting. These results suggest that lo-fi is an effective alternative to standard multi-node fine-tuning with communication.

### 3.4 How much is the speed-up, really?

We have shown that lo-fi produces high accuracy models without communication during fine-tuning. This leads to an important practical question: what is the wall-clock advantage of eliminating communication between nodes during fine-tuning? We examine the wall-clock training time advantage once nodes are allocated and also the time it takes for node allocation on a slurm cluster. Note that these experiments are for the DeiT-III (Touvron et al., 2022) models in the image classification setting.

**Wall-clock advantage.** To examine the wall-clock advantage of lo-fi compared to the baseline we use A100 GPUs on AWS with fast interconnect of 400 GBps (EFA). This is representative of a fast and modern large scale neural network training set-up. In particular, we want to understand the effect of using modern distributed training tools, and also varying batch size. We note that our results depend critically on the quality of the interconnect between nodes. In a setting with a slower interconnect such as standard ethernet, we would expect the training speed-ups to be more substantial. In a setting with a faster interconnect such as TPUs, the training speed-ups should be more minor.

A recent innovation in distributed training tooling is to overlap the backwards pass computation and gradient communication—the gradients for layer $\ell - 1$ can be computed at the same time as communicating the gradients for layer $\ell$ (Li et al., 2020a; Paszke et al., 2019) [4]. We experiment with turning on and off this overlapping communication/computation feature, finding substantial reductions in communication overhead when overlapping communication and computation. We also experiment with changing the batch size. In general, we observed that when using a smaller batch size, communication will account for a larger portion of training time. This is because the size of the gradient does not depend on the batch size, so absolute communication cost does not depend on batch size. However, using a smaller batch size will lower the total computation time and therefore communication cost will account for a larger fraction of the total training time.

---

[4]Overlapping communication/computation on by default in PyTorch $\geq$1.5 (Paszke et al., 2019; Li et al., 2020a).

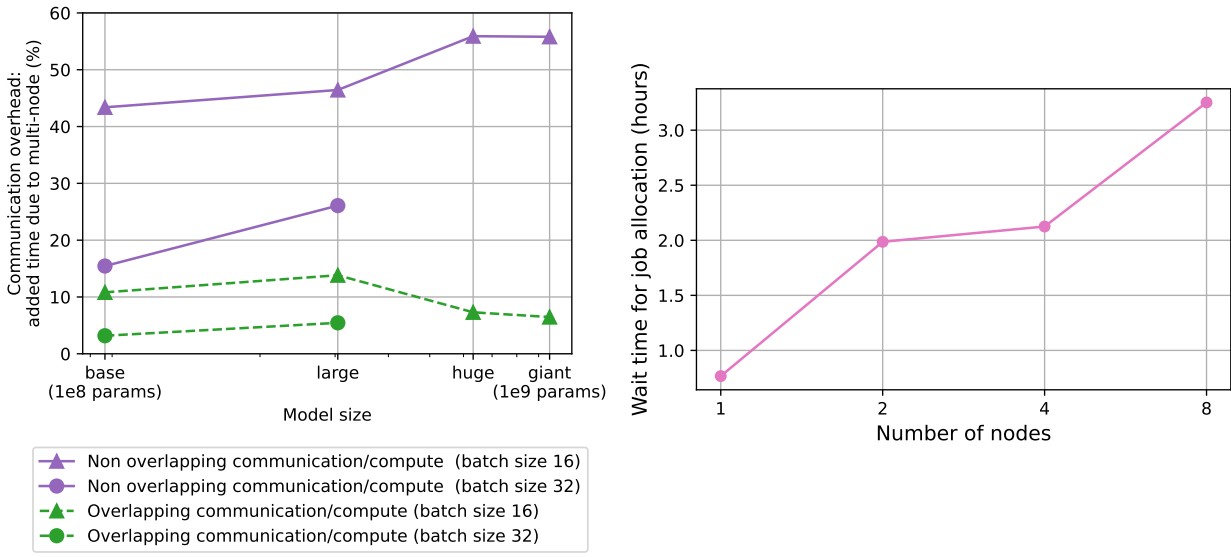

Figure 7: **(Left)** On an AWS cluster we show on the $y$-axis the wall-clock overhead observed when switching from 1 to 4 nodes using models from the DeiT-III repository (Touvron et al., 2022) and constant per-GPU batch size. 100% indicates that the job becomes twice as slow while 0% indicates no difference switching from 1 to 4 nodes. With the method of overlapping the communication and computation in the backward pass (Li et al., 2020a), the slow-down is less substantial than we initially expected, especially for larger per-GPU batch sizes. The huge and giant models are deeper and there is more opportunity to overlap communication and computation. **(Right)** Jobs requiring only one node schedule faster than jobs requiring four nodes on the slurm cluster that we use for these experiments. This plot shows the median per-day wait time averaged over three months of job data on this cluster.

Our experiments with varying batch size and turning on and off overlapping communication/computation are presented in Figure 7 (left). These experiments are for the vision transformer models DeiT-base, DeiT-large, DeiT-huge and DeiT-giant, ranging from roughly $10^8$ to $10^9$ parameters. On the $x$-axis we show the different model sizes, while the $y$-axis shows the additional wall-clock time required to go from a 1 node to 4 node job (i.e., 0% indicates that the 4 node job is the same speed as the 1 node job, while 100% indicates that the 4 node job is twice as slow). In this experiment, the number of iterations and batch size per device is fixed[5]. We found that without overlapping communication/compute, shifting to a multi-node settings results in a substantial increase in training time of 25-55% (purple lines). However, overlapping communication and compute has proven surprisingly effective, reducing the communication cost to <10%.

A potential issue with these experiments is that they currently reflects a "state-of-the-art" cluster setting, and actual workflows may be slower. We also believe that both GPU memory size and network bandwidth will improve in the future. Higher GPU memory capacity will allow users to train larger models, resulting in higher communication overhead, while higher network bandwidth will help to reduce the communication delay.

Finally, we note that lo-fi can help the straggler and jitter problem in distributed training, where different nodes might experience random slowdowns due to various reasons. In standard data-parallel, synchronization will take place multiple times per iteration, such that any random slowdown on any node will slow down the entire run. Since lo-fi needs only one communication at the end (which can even be asynchronous), the straggler/jitter problem is no longer an issue.

---

[5]We note that scaling with fixed batch size may be unrealistic for certain problems as large batch sizes can cause accuracy to drop, which would be a reason to use lo-fi.

**Scheduling advantage.** For modern cluster workloads, both on private and public clusters, the wait time to schedule a job can increase the total training time, especially during periods of heavy cluster usage. Since single-node jobs require fewer simultaneous resources to run, they should schedule faster, reducing the total training time. To measure this, we analyzed the time required to schedule a 1-node job vs. a multi-node job on a large slurm-based cluster and present the results in Figure 7 (right). These wait times are averaged over all jobs run on this cluster over a three month period. We found that scheduling a single node job was notably faster than multi-node jobs, taking ∼45 minutes for 1 node, ∼2 hours for 2-4 nodes, and ∼3 hours for 8 nodes.

We note that these results are specific to the cluster used in these experiments and may or may not be representative of other clusters depending on their scheduling algorithm and workload distribution, amongst other factors. We also note that the scheduling benefit will only apply when using implementation B in which each group is trained independently (as described in Section 2). Regardless, we thought that it may be useful to collect and present this empirical data, providing quantitative support for the observation from (Li et al., 2022) that jobs requiring fewer nodes schedule faster.

### 3.5 Does jointly training to increase diversity across groups improve lo-fi performance?

Previous work from (Gontijo-Lopes et al., 2021; Wortsman et al., 2022) has shown that more diverse models trained with different hyperparameters produce larger benefits when ensembles or weight averaged and also (Li et al., 2022) which showed that ensembling or weight averaging specialists trained on different domains incurs the largest benefit. We therefore asked whether encouraging diversity automatically through regularization during training might improve the performance of the final lo-fi model.

While this strategy did indeed produce models with a larger averaging benefit (avg. model - best individual model), it also decreased the accuracy of the individual models such that overall performance was the same or worse than simply training the lo-fi components independently. We also tried pulling together the predictions of the models, which is also known as co-distillation (Anil et al., 2018; Sodhani et al., 2020). This improved the accuracy of the individual models, but as model diversity decreased, the benefit from weight-averaging was reduced, also leading to overall lower accuracy. We explored a number of variations of these approaches which we discuss in more detail in Appendix A.

## 4 Related work

**Averaging and linearly interpolating models.** Averaging or interpolating the weights of neural networks is a common technique for improving accuracy.

Weight-averaging techniques for optimization date back to early work in convex optimization (Ruppert, 1988; Polyak, 1990). In deep learning, an exponential moving average (EMA) of weights can be used to improve accuracy (Szegedy et al., 2016). Another popular approach is Stochastic Weight Averaging (SWA) (Izmailov et al., 2018) which uses a uniform average of weights saved at each epoch while training with a constant or cyclic learning rate. Indeed, the SWA method was motivated in part by the analogy between weight-averaging and ensembling.

While SWA and EMA average weights along the training trajectory, there has also been substantial interest in averaging weights across independent trajectories. In particular, Nagarajan & Kolter (Nagarajan & Kolter, 2019) observe that the weights of two models that are fine-tuned independently on MNIST (LeCun et al., 2010) from a shared initialization can be interpolated without increasing loss. For more difficult problems such as ImageNet, this naive linear interpolation encounters a high error barrier (Frankle et al., 2020; Fort et al., 2020). However, Frankle et al. (2020) observe that when the first part of the optimization trajectory is shared and the remainder of training is independent, models can once again be interpolated without reducing accuracy. They refer to this phenomena—interpolating weights without accuracy loss—as *linear mode connectivity*. Neyshabur et al. (2020) observed a similar phenomenon when interpolating between model pairs that are *fine-tuned* from a shared initialization on a new task. This observation was extended to interpolation between a zero-shot model and fine-tuned model with the WiSE-FT approach (Wortsman et al., 2021), to many models fine-tuned with different hyperparameters with model soups (Wortsman et al.,

2022), to models fine-tuned on different datasets with Ilharco et al. (2022), and for creating better pre-trained models by Choshen et al. (2022). The overall observation that the objective landscape can appears roughly convex was also made by Li et al. (2018). While all of the aforementioned weight-averaging employ simple linear interpolation, more advanced weight-averaging techniques have also been developed with promising results (Matena & Raffel, 2021).

Recently, Li et al. (2022) introduced branch-train-merge which is at the intersection of model combination and distributed training. They consider the case where the training data is partitioned into different textual domains, then train an individual expert model for each domain. As they are training from scratch, they first require an initial seed phase. They then combine all of these experts via weight averaging or ensembling to outperform the dense baseline of training one large model on all of the data. The main difference are that our work is for fine-tuning, and we do not assume the data is partitioned into different domains.

Other research in the area includes Garipov et al. (2018) and Draxler et al. (2018) who concurrently found that two neural network solutions trained independently can be connected by a simple curve along which loss remains low. These findings were generalized by Benton et al. (2021) who learn high dimensional low-loss connectors between individual solutions. Concurrent work with Benton et al. (2021), Wortsman et al. learned these high dimensional low-loss subspaces from scratch. Then, Entezari et al. (2021) conjectured that all solutions could be made to be linearly connected by applying a permutation to the weights which does not change the function. Ainsworth et al. (2022) recently made progress towards confirming this conjecture. However, unlike the model interpolations we observe here, and have previously been observed (Wortsman et al., 2022; Li et al., 2022), the interpolations in Ainsworth et al. (2022) so far do not improve models in terms of accuracy. Regardless, they are interesting from a scientific perspective, and suggest the possibility of applying methods such as lo-fi for training from scratch in the future, although there is currently no evidence towards this.

**Distributed training and fine-tuning.** Distributed training (Li et al., 2020a; Yuan et al., 2022) and fine-tuning (Borzunov et al., 2022; Wang et al., 2022) are increasingly important in deep learning as models become larger.

An overview of the many standard approaches is detailed by Weng & Brockman (Weng & Brockman, 2022), including i) data-parallelism, where data is split among devices, ii) pipeline parallelism, where different layers are split among devices, and iii) tensor parallelism, where individual layers are split among devices. We note that these approaches are not mutually exclusive. Indeed, one can use pipeline parallelism to distribute a model across a node, then use data-parallelism across nodes. lo-fi is proposing an alternative to data-parallelism across nodes—instead of synchronizing the updates between nodes during fine-tuning, each node independently produces a model which is averaged at the end of fine-tuning. We emphasize that lo-fi can still be used if there is pipeline parallelism across the node.

There have previously been many alternatives proposed to synchronizing gradients each step. The idea of training several models in parallel and averaging their weights once at the end of training has been investigated at least since (McDonald et al., 2009; Zinkevich et al., 2010). The focus in those works is on convex models and training from scratch, rather than fine-tuning. While lo-fi is simply borrowing these techniques from convex optimization and applying them to fine-tuning for deep learning, we believe that these findings are interesting and useful from a practical standpoint.

Another alternative, HogWild (Recht et al., 2011) proposes asynchronous communication. The difference between HogWild and lo-fi is that lo-fi never communicates during fine-tuning, so it's as if the hogs each have their own individual farm. As another alternative, local-sgd (Stich, 2018; Ortiz et al., 2021) communicates updates every $k$ steps instead of every step. lo-fi is equivalent to local-sgd applied to fine-tuning where $k$ is the number of fine-tuning epochs.

There have also been compelling recent methods for more efficient and accessible pipeline or tensor parallelism to enable learning or inference with extremely large models. For instance, with Petals (Borzunov et al., 2022) certain layers of very large models are computed and communicated among coordinated users. Also, researchers have been using decentralized training for very large models (Huang et al., 2019; Yuan et al., 2022) which is made possible by, e.g., compressing communication (Wang et al., 2022; Kairouz et al., 2021;

Xie et al., 2020; Faghri et al., 2020; Li et al., 2020b). Indeed, even inference with very many-billion parameter models can pose interesting challenges (Dettmers et al., 2022). Our approach is orthogonal to the work in compressing communication, as we are instead removing communication, but may prove useful for large-scale decentralized training.

There is also the active research area of federated learning (e.g., Kairouz et al. (2021); Pillutla et al. (2022)), which has recently been explored in transfer settings (Nguyen et al., 2022). In federated learning, the data on each client is different and updates are usually communicated every $k$ steps. While lo-fi only considers the easier setting of IID data, it is possible that similar approaches based on weight averaging to reduce communication may prove beneficial for privacy-preserving machine learning.

## 5 Limitations and conclusion

**Limitations.** There are many limitations discussed throughout this text. For instance, we found that when fine-tuning CLIP ViT-L on ImageNet and WILDS, lo-fi needs to observe more data to exceed the baseline. This is similarly true during language model fine-tuning (Section 3.3). Therefore, we most recommend lo-fi when communication costs are prohibitive. A final limitation is that lo-fi can only achieve matching accuracy when no new parameters are introduced, which we accomplish with a "zero-shot" initialization, or via LP-FT (this does not come up during language model fine-tuning).

**Conclusion.** Overall we have observed that communication between nodes is not required during fine-tuning in certain settings. These findings may prove beneficial to a number of settings including large-scale decentralized fine-tuning and privacy-preserving ML and represent a promising step in the overall direction of developing models like open-source software (Raffel, 2021) in which many institutions can collaboratively fine-tune a large model if none has the resource to do so individually. As more workloads shift to fine-tuning of pre-trained models and models grow increasingly larger, we hope that our results will help to reduce barriers to large-scale models.

### Acknowledgements

We thank Beidi Chen, Surya Ganguli, Caleb Ho, Gabriel Ilharco, Teng Li, Mansheej Paul, Alex G, Andrew Saxe, David Schwab, Shubho Sengupta, and Hugo Touvron for useful discussions.

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

# A Negative results when using regularization to promote model diversity

We also experiment with explicitly regularizing the models to have diverse predictions. We are motivated by previous work which shows that having diverse models can lead to better ensembles or weight-averages (Brazowski & Schneidman, 2020; Gontijo-Lopes et al., 2021; Wortsman et al., 2022; Li et al., 2022). In particular, we wanted to push apart model predictions from different nodes. However, in the standard set-up, different nodes observe different data. As an alternative to this, at each iteration we had pairs of nodes observe the same data. We did this without reducing batch size by using cutmix (Yun et al., 2019) with the data from one node to another. However, without any further changes, this modification reduced accuray by roughly 0.15pp on ImageNet. We suspect this is because the models on different nodes became less diverse by sharing data.

Let $y_1$ and $y_2$ be the predictions from two nodes who observed the same data. We experimented with adding the term $\lambda \cdot \mathcal{D}_{\mathrm{KL}}(y_1 \| y_2)$ to the loss where $\mathcal{D}_{\mathrm{KL}}$ is the KL divergence. The motivation was that with $\lambda < 0$ the models would become more diverse as in (Brazowski & Schneidman, 2020). When $\lambda > 0$, this procedure would become co-distillation (Anil et al., 2018) and hopefully accelerate training. However, while $\lambda < 0$ improved the absolute benefit from averaging models, it reduced the performance of the individual models. Moreover, while $\lambda > 0$ improved the performance of the individual models, it decreased the benefit of averaging. These experiments are presented in Figure 8, finding that any change to $\lambda$ lowered the accuracy of lo-fi. We also experimented with a number of other approaches to improve diversity, including using other distance metrics, only pushing apart incorrect examples, pushing apart model weights, and also taking the PCA of the predictions and pushing apart in the unimportant PCs. However, all of these approaches exhibited the same qualitative effect: improving diversity reduced individual model performance and increased the benefit of averaging, but not by enough to offset the individual model reduction. Interestingly, while our search was not exhaustive, this result may suggest that simply fine-tuning models with random seeds might produce the optimal amount of diversity for ensembling and model averaging.

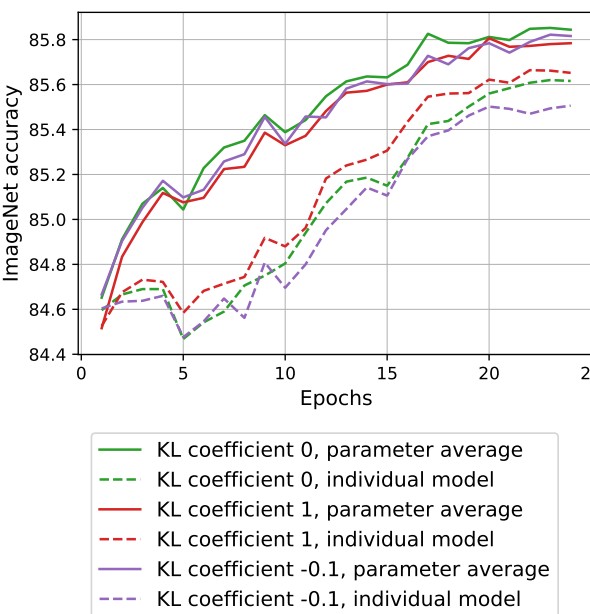

Figure 8: We tried "pushing apart" or "pulling together" the models at each of the nodes by adding the KL divergence between their predictions to the loss. Using a positive coefficient is equivalent to co-distillation (Anil et al., 2018) which improved the individual models but decreased the accuracy of the average. Using a negative coefficient as in (Brazowski & Schneidman, 2020) overall increased the improvement from averaging but decreased the accuracy of the individual models. Overall, these approaches did not improve lo-fi.

Figure 9: We supplement Figure 2 with the additional baseline of standard multi-node fine-tuning with communication. We do not replace Figure 2 as the intention of Figure 2 is to visualize how the curve changes as the $x$-axis changes, and our intention is not for readers to view the figure as a comparison between lo-fi and the baseline.

## B  A comment on learning rate in Figure 1

A question emerges when examining Figure 1: why does the accuracy of the individual node first dip before coming back up? The answer is due to learning rate. While both lo-fi and the baseline use the same learning rate, which is the learning rate used by DeiT-III paper of 3e-4, the individual lo-fi nodes have a smaller global batch. Therefore, this same learning rate acts larger. We tried to increase the learning rate for the baseline so that it also had this down-then-up trend but it resulted in worse accuracy. We also tried changing the learning rate for lo-fi but this also reduced accuracy. This is similar to an observation made in the context of EMA in model soups (Wortsman et al., 2022), which is that the best model for averaging weights is not necessarily the best model overall. We believe the learning helps individual nodes produce models which are different, and there is therefore more benefit from their combination.

## C  Additional experimental results

This section presents additional experimental results. First, we supplement Figure 2 with the additional baseline of multi-node training (Section C.1). Next, explore the application of lo-fi for fine-tuning on tasks from GLUE Wang et al. (2018).

### C.1  Adding the additional baseline of multi-node training to Figure 2

In Figure 9 we supplement Figure 2 with the additional baseline of multi-node training. Due to computational constraints for simultaneous nodes, we do not exceed 12 nodes. Overall we observe similar curve shapes for both the baseline and lo-fi.

### C.2  Additional natural language processing results

In Section 3.3 we explored fine-tuning with the language modeling objective. This was because standard fine-tuning tasks in natural language processing are relatively small and don't require multiple nodes. In this section, we expand our results to also consider fine-tuning a RoBERTa-base (Liu et al., 2019) model on tasks from the GLUE benchmark (Wang et al., 2018). Instead of fine-tuning on multiple nodes without communication, we instead fine-tuning on multiple GPUs without communication. Since we are introducing new parameters (i.e., the classification layer), we use LP-FT similarly to the experiments in Section 3.1. We accomplish this via a common first epoch among lo-fi workers. The baseline is then 40 epochs with communication among 4 GPUs while each of the 4 lo-fi workers is 10 epochs on 1 GPU before a final averaging step. In contrast to vision experiments, NLP fine-tuning can sometimes have low accuracy depending on the seed (Dodge et al., 2020). To account for this we use the greedy averaging technique from Wortsman et al. (2022) to combine models instead of the uniform averaging used throughout the rest of the paper.

|          | SST2      | COLA      | QQP       | MRPC      | RTE       | MultiNLI  |
|----------|-----------|-----------|-----------|-----------|-----------|-----------|
| baseline | 94.30     | 61.34     | **89.29** | 92.10     | **77.98** | 87.51     |
| lo-fi    | **95.30** | **63.33** | 89.12     | **92.20** | **77.98** | **87.83** |

Table 4: The performance of lo-fi compared to the baseline when fine-tuning on GLUE tasks (Wang et al., 2018). Details in Section C.2.

