# OpenReview forum: "lo-fi: distributed fine-tuning without communication"
_TMLR — Accepted by TMLR_

### Review · Reviewer_Q9VY · 2022-11-19

**Summary Of Contributions:**

This paper demonstrates that fine-tuning a model on individual nodes and then averaging their weights afterwards can match (or even outperform) training one model on all the nodes using data parallelism. They demonstrate that the results hold when fine-tuning vision and language models.

**Audience:**

Yes

**Claims And Evidence:**

Yes

**Requested Changes:**

- Simply strengthen work: NLP experiment which fine-tune a LM on some standard downstream task for fine-tuning.
- Simply strengthen work: A baseline in figure 2 which uses gradient communication across various number of nodes. This would indicate if using gradient communication experiences diminishing returns after 4 nodes.


**Strengths And Weaknesses:**

Strengths
- Strong experiments across different modalities that verify their claim
- Additional experiment on wall-clock time demonstrates the actual use-case of their method
- Good ablations on OOD data and effect of number of nodes

Weaknesses
- The NLP experiments fine-tune OPT on CommonCrawl, which is normally used for pre-training, not fine-tuning. This difference matters since pre-training is done on new parameters and fine-tuning is done on non-new parameters and the behavior of lo-fi is different in these two setups. “lo-fi can only achieve matching accuracy when no new parameters are introduced” - limitations.

---

> ### Author Response · Authors · 2022-12-09
> **Response to Reviewer Q9VY**
>
> Thank you for the thorough and constructive review, we hope that this response addresses your concerns. Overall, we have included the suggested experiments in Section C which we believe improves the paper and thank you for these suggestions.
>
> First, we expand our experimental results for NLP fine-tuning to include fine-tuning a roberta-base model on tasks from the GLUE benchmark. The initial reason for choosing a language modeling task is that it requires multiple nodes, whereas other tasks used for fine-tuning are relatively smaller. Therefore, instead of fine-tuning on multiple nodes without communication, we instead fine-tuning on multiple GPUs without communication. Our results are in Section C.2, Table 4 of the Appendix – we find that lo-fi can match or outperform the baseline on 5 of the 6 tasks we tried.
>
> We’ve also added the requested baseline to Figure 2 (see Section C.1 and Figure 9 of the Appendix). Indeed, the trend for standard does follow lo-fi. Due to computational constraints we were unable to run with over 12 nodes simultaneously.

---

> > ### Comment · Reviewer_Q9VY · 2022-12-13
> > **Response to authors**
> >
> > Thank you for running the additional experiments. Overall, my concerns are addressed.

---

### Review · Reviewer_iWNq · 2022-11-22

**Summary Of Contributions:**

- This paper proposes a method lo-fi, which aims at achieving distributed fine-tuning without extra communication. In its vanilla version, lo-fi simply conducts local fine-tuning on each computing device before a final averaging step.
- Motivations and intuitions are provided for the proposed lo-fi method.
- Empirical and ablation studies were conducted to justify the effectiveness and communication efficiency of lo-fi.

**Audience:**

Yes

**Broader Impact Concerns:**

I do not find any concerns about the ethical implications of the work. Thus I would not urge the authors to add a Broader Impact Statement.

**Claims And Evidence:**

Yes

**Requested Changes:**

- One interesting direction/study that can be added to improve the current draft is: [1] is guaranteed to converge for convex functions. We all know that the loss landscape of a deep neural network is not convex at all, however why PSGD works particularly well for fine-tuned deep neural networks? Does the pre-training make the loss landscape more convex? I believe that some studies in [2] can be borrowed.
- The proposed method lacks novelty. The authors have to convince potential readers very well why their proposed method is important.
- The authors are encouraged to compare lo-fi against the other efficient fine-tuning methods, e.g., LoRA and Adapter.
- A discussion on the compatibility of lo-fi to model and pipeline parallelism can improve the quality of the paper.
- The writing quality of the current draft can be improved.

[1] https://papers.nips.cc/paper/2010/file/abea47ba24142ed16b7d8fbf2c740e0d-Paper.pdf
[2] https://arxiv.org/abs/1712.09913

**Strengths And Weaknesses:**

Strengths:
- The idea of lo-fi is easy to follow.
- Improving the communication efficiency of distributed fine-tuning is a promising research direction.

Weaknesses:
My major complaint about this paper is that the idea is simply too straightforward. Conducting distributed computation with only final gradient/model aggregation (i.e., after the convergence of all locally trained) is a simple and old idea (please see PSGD proposed in [1], which appeared at NeurIPS 2010). Lo-fi, in my view, is a simple extension of PSGD to fine-tuning tasks, which has limited novelty.

More concerns are summarized below:
- Many methods have been developed to make fine-tuning very memory and/or computation-efficient, e.g., the ones in [2-3]. In fact, those methods also manage to save communication costs under a distributed scenario (as their number of active tunable parameters is so few). I wonder how lo-fi compares against LoRA and Adapter in [2-3].
- The proposed lo-fi method seems to only apply to data parallelism. For the model that can not fit into a single GPU memory, how does lo-fi work? In another word, can lo-fi be modified to support model and pipeline parallelism strategies?
- The draft seems to be written in rush, there are a few issues in formatting/writing, e.g., there is an abrupt line break at the end of the second paragraph of the Introduction.

[1] https://papers.nips.cc/paper/2010/file/abea47ba24142ed16b7d8fbf2c740e0d-Paper.pdf
[2] https://arxiv.org/abs/2106.09685
[3] https://arxiv.org/pdf/1902.00751.pdf
[4] https://cs.stanford.edu/people/matei/papers/2021/sc_megatron_lm.pdf
[5] https://arxiv.org/abs/2201.12023

---

> ### Author Response · Authors · 2022-12-09
> **Response to Reviewer iWNq**
>
> We thank the reviewer for their review. Thank you also for indicating that the claims made in this paper are supported by accurate, convincing, and clear evidence, and that some individuals in TMLR’s audience would be interested in the findings of this paper. We appreciate the reviewer’s time and hope that the following response addresses concerns.
>
> > My major complaint about this paper is that the idea is simply too straightforward.
>
> Overall, we agree which is why we’ve selected TMLR for submission which has the following on their acceptance criteria page (https://www.jmlr.org/tmlr/acceptance-criteria.html): “Nor should it form the basis for rejecting work on a method considered not “novel enough”, as novelty of the studied method is not a necessary criteria for acceptance.”
>
> > please see PSGD proposed in [1], which appeared at NeurIPS 2010
>
> While our submission already does cite PSGD, we have updated the draft to make the relation more clear (third paragraph of the 'distributed training and fine-tuning' section of the related work). Indeed, lo-fi is simply borrowing these techniques from convex optimization and applying them to fine-tuning for deep learning. Nonetheless, we believe that these findings are interesting and useful from a practical standpoint.
>
> > The proposed lo-fi method seems to only apply to data parallelism. For the model that can not fit into a single GPU memory, how does lo-fi work? In another word, can lo-fi be modified to support model and pipeline parallelism strategies?
>
> With the exception of Table 3, all experiments in our submission remove communication between nodes but allow communication between the GPUs on each node. If the models do not fit on the GPU they can be distributed across the GPUs on the device. We do not currently have a solution for models which don’t fit on a node, but we believe this is a very interesting direction for future investigation.
>
> > Does the pre-training make the loss landscape more convex? I believe that some studies in [2] can be borrowed.
>
> This is a great point. Yes, this is roughly our hypothesis for why lo-fi works. We state in the introduction that: “one notable difference between pre-training and fine-tuning is that fine-tuned models appear to lie in a single low-error region (Neyshabur et al., 2020)”. We have updated the related work to include the suggested reference (third paragraph of related work).
>
> > The authors are encouraged to compare lo-fi against the other efficient fine-tuning methods, e.g., LoRA and Adapter.
>
> Thank you for the suggestion. Overall, we view lo-fi as complementary to efficient fine-tuning methods. One could still use lo-fi when using an efficient fine-tuning method.

---

### Review · Reviewer_xXhU · 2022-12-04

**Summary Of Contributions:**

The work presents a distributed way of finetuning by finetuning separately and ensembling\merging models, with little communication. This allows for different nodes (groups of processing units that are interconnected well) not to communicate during training and still perform as well or even better. A crucial matter here is that, per node, the amount of computation is held constant. So it produces a protocol where training is more efficient when node communication is slow or problematic.

**Audience:**

Yes

**Claims And Evidence:**

Yes

**Requested Changes:**

Mainly, things in the framing bother me: the relations to ensembling and individual training and the over discussion of the theoretical reasons to believe it would work (which are great but do not replace talking about this work too...), but under discussion of motivation and terminology.

The explanation of the motivation is unclear to me. Also some of the nuances of the experimental setup, specifically those related to amount of compute, the answer to the question what really differs between each baseline (except the hardware speed and other factors that do not affect the reported results), especially between the one node lo-fi and the regular training. I believe I understand them now, but it was not explicit in any one place.

Minor adjustments are just the minor weaknesses,

**Strengths And Weaknesses:**

Strengths:
The paper has a lot of substance, experiments and ablations.
The results show a clear trend, there is little doubt the method works and achieves the claims in the paper.

Minor:
The robustness to more epochs (e.g. Fig 4,5) is interesting and a good property. Have any guess as to why is it happening? It is also a small weakness that it sometimes requires more epochs to surpass the baseline, is that a cost of distributing or related to the drop rate?

Weakness:
While the framing discusses distributed finetuning, this fits the results only with some caveats. Finetuning can be done without multiple GPUs and reach reasonable results and this method could be seen as a way of ensembling weak models. This is little discussed, unless I missed it, but we see that training on just one node works not that bad, so essentially what the paper proposes is training without distribution, stopping before convergence (right?) and getting a lower result, and then performing a model-soup-like ensemble that, as a good ensemble, improves results over the individual models. So in essence we could just put more computing effort, calculate anything we want more times (for example the full fine-tuning or a partial one) and make an ensemble to get better results. That is far from new or surprising. So the paper does not really say deep things about the low communication, but instead could be understood as saying things on ensembling of undertrained models, which we can expect. This alternate view should be better discussed. For example, why train with a small batch size (1/K) and not regularly (accumulate to account for missing nodes) but stop in the middle of training, is it better? worse? faster? Ensembling weak models also goes a lot back, connecting this paper to that branch may allow others to more easily improve it.

It should be emphasized more how lo-fi is in comparison to regular non-distributed training. Overall data is the same, overall GPU number is the same (although don't have to be), what about logical batch size (after accumulation), per node data, overall computation time? Except for the communication what changes with it? There are 3 aspects (data, compute and accuracy) and each of them is separated (device\node\overall + maybe accumulated batches). It takes time to understand, what of those require further costs, what are the possible gains and what is held constant. The accuracy part is quite elaborated, but the other aspects are spread across the paper despite being the real reason for someone to prefer this method.

As far as I can tell (and the fact that I am unsure is discussed above), the difference between lo-fi one node and regular training is that lo-fi batches are 1/K size, but same learning rate. In that case, the authors show that training was worse when choosing the wrong learning rate, there is even Appendix showing it is likely the case. This raises suspicions, could the one node perform ok with an ok learning rate? (with gradient accumulation, surely it can because logically it is equivalent, the paper should mention it and explain why this is possible but not advisable as it is slow) If it can, then no problem, more probable is that we will see that the method can just boost some lower-performing models by fusing them (like model soups etc.). But then, the framing is different, we want to just train faster, not to get better results, because better results are achievable by training several times and merging (as we said model soups). This method proposes to train inefficiently without communication and not lose (much?), but gain in speed. So more emphasis should be put on times. Specifically, they can train each node better (fit the parameters, you made so much effort fitting the baselines well you can fit your own too), and stop the training given the same computation as given to the other method but not communication, or even stop before, and show at which point you can stop. Note, running until convergence will tell us nothing, the framing here is about no communication, but without communication and with more compute\time anything could be done, without Lo-Fi, just train on one node K times longer and accumulate gradients, this will be performing the computation of the K nodes on 1, and is not interesting.


Minor:

The introduction mentions nodes, but the method defines devices. Are those the same? Do those do anything other than processing? would they be called processing units (GPU TPU CPU?)? Device for me hints of phones, tablets, laptops, PCs etc. but, it may be just me, and I'm not even a native speaker. Only when discussing some devices talking to each other and others that do not did I understand you are referring to slurm-like definitions. You should define those somewhere. A node is a group of devices which can better communicate within themselves than with devices from different nodes. Also, make sure to define things in a general manner, if hardware would change tomorrow and call a group for example "pod" then the methodology of your paper, that in no way relies on the hardware should still be clear. Figure 1 is actually clearer in describing the setting.

It is not overflowing the paper, so it is not critical, but still, because no experimental setup is provided, various details are written in places where they give no benefit for the reader. For example, when trying to understand the method, "In the majority of our experiments, each group is a single node containing 8 GPU devices." is of no interest.  Or which code is used for DeiT which is mentioned at least 5 times.

The introduction describes regions in the low space in quite a detail, which to me seems to miss the point. The motivation for the work is not clear. This is only the background to why it is possible or what came before this work. For example, what is a node (first mentioned when the method is introduced), or why would costs of communication between nodes (GPUs) would ever be "prohibitive". This is the motivation of this paper, communication during fine-tuning, not low regions in space, which are related but do not make this paper exciting, just possible. There is another possibility, and that is that the paper does make regions in space during finetuning its main motivation, but this would require reframing if not a bit different focus on the experiments. While changing this will mean rewriting the introduction, I don't see it as necessary for the paper to be published, just for it to be more exciting or convincing.

Why did you use different implementations in text and vision experiments? Did you test how they affect results or times? I also could not follow the explanation of the implementations, you state this is implementation after the one-time communication, why batches and gradients are still needed at this stage?

Figure 1 which the paper heavily relies on does it some disservice. The "lo-fo(individual node)" has little to do with lo-fi. It is not that the paper proposes to use this method. This looks like there are two things the paper suggests, and one of them might be tolerable despite lower performance.

If every node can already train the model by itself, and you allow only one communication, why not have a small finishing touch just to improve over the noise of averaging\merging\fusing the model? For example, only learn the last layer to make sure it predicts based on the inputs. Or perform some final batches after the fuse (we have found it to help in other scenarios).

The tables would benefit from a split to in domain and out of domain, as this is the main claim and the names of individual test sets are necessary but not the main message. This would highlight where lo-fi "wins".

When you write "drop" in Table 2 you mean stochastic depth drop, right? worth writing it in the caption, as drop with little context may be dropout or other things.

Why is appendix B the first one referenced in the text? Makes you think you skipped a note somewhere before. (By the way, the addition of this mystery is very nice)

"we found it was necessary for achievingmatching performance with lo-fi." it is quite ok not to achieve the same performance with a more restricted method. how big was the drop?

"in which this the benefits" typo

---

> ### Author Response · Authors · 2022-12-09
> **Response to Reviewer xXhU**
>
> Thank you for the very thorough review, we feel as though your suggestions greatly improve the paper quality.
>
> Framing & motivation weakness:
>
> Thank you for bringing this concern to our attention. We would like to address this concern and appreciate your feedback above on how to do so.
>
> One thing we want to emphasize is that the experimental setup and methodology was grounded in experiments that people were already running (our main experiment uses an identical set up to DeiT-III in21k fine-tuning). In other words, we took previously established fine-tuning paradigms which had communication between nodes, and showed that lo-fi can match performance without communication. We believe that as models and datasets get larger, fine-tuning on a single node may simply become too slow for some problems compared to multi-node training. Our main motivation here was to provide even more of a speed up than standard multi-node training without the requirement that nodes be connected by fast interconnect.
>
> We agree that with gradient accumulation, these distributed runs could have been run on one node. However, they would then be roughly k times slower (where k is the number of nodes). With lo-fi, there is no such slow down; rather, it is possible to speed up fine-tuning instead. We emphasize that at each point on the x-axis of Figure 1, 4, 5, and 6, as well as for all models in Tables 1 and 2, the number of floating point operations is fixed (other than those associated with communication).
>
> We agree that the results are unsurprising from the perspective that ensembling weak models is known to provide improvements. However, we are ensembling the weights of the models, so there is no added cost during inference, and this is less well studied by comparison. Since neural networks contain many non-linearities, it is initially surprising that averaging weights works at all. Moreover, the suggestion of simply simulating lo-fi on a single node is quite an interesting direction for future work.
>
> We also want to emphasize that the baseline is exhaustively hyperparameter tuned. Indeed, the fine-tuning runs outperform the numbers reported in the DeiT-III paper because of this exhaustive hyperparameter tuning for the baseline (Table 1). We were surprised that this same learning rate worked off-the-shelf for lo-fi, and the appendix section is on experiments where we tried different learning rates for lo-fi which we initially expected to help but did not. We hope this removes your hesitations.
>
> Minor:
>
> 1.
>
> > The robustness to more epochs (e.g. Fig 4,5) is interesting and a good property. Have any guess as to why is it happening?
>
> Thanks for the interesting question. If we had to guess, we would say that it is due to less overfitting. This is because, when fine-tuning for 24 epochs, each node in 4-way lo-fi is effectively only fine-tuning for 6 epochs. As in standard training, data is partitioned between nodes at each epoch. But unlike standard training, there is no communication so each node “sees” less data (even though the aggregate amount of samples seen is exactly matched between lo-fi and the baseline). This answer is likely not the full story and deserves further investigation in future work.
>
> 2. Overall clarity (e.g., devices vs. nodes)
>
> We apologize for the issues in clarity that you highlight. By device we mean either GPU or TPU, and our experiments are on nodes which contain 8 GPUs.
>
> 3.
>
> > Why did you use different implementations in text and vision experiments?
>
> Because of differences in code bases. We used huggingface for the NLP experiments, so it was more convenient to use a slightly different implementation. We tested both implementations for the DeiT-III fine-tuning experiment and found negligible differences. Overall we wanted readers to be aware there are two possible ways to implement the algorithm.
>
> 4.
>
> > Figure 1 which the paper heavily relies on does it some disservice. The "lo-fo(individual node)" has little to do with lo-fi.
> Thanks for this interesting perspective. This individual node performance provides a useful reference point, but we agree that prefacing it with “lo-fi” does some disservice, so we’ve removed that in the revision.
>
> 5.
>
> > If every node can already train the model by itself, and you allow only one communication, why not have a small finishing touch just to improve over the noise of averaging\merging\fusing the model?
>
> This is an interesting idea. Indeed, we reference Choshen et al., 2022 who do additional training after averaging and show this helps. There is also the possibility of using more sophisticated averaging schemes such as Fisher-Weighted (https://arxiv.org/abs/2111.09832). We believe these experiments are beyond the scope of the paper but are interesting suggestions for future work.

---

> > ### Author Response · Authors · 2022-12-09
> > **Response to Reviewer xXhU (part 2)**
> >
> > 6.
> >
> > > When you write "drop" in Table 2 you mean stochastic depth drop, right?
> >
> > Yes, thank you we’ve fixed this and the typos that you’ve caught, thank you very much.
> >
> > 7.
> >
> > > "we found it was necessary for achieving matching performance with lo-fi." it is quite ok not to achieve the same performance with a more restricted method. how big was the drop?
> >
> > We found that performance dropped to similar levels as  the individual node models, i.e., averaging did not have the desired ensemble like effects.

---

### Author Response · Authors · 2022-12-09
**Response to all reviewers**

We thank the reviewers for their comprehensive and constructive reviews. We hope that concerns are addressed, and believe that the suggestions have greatly improved the paper. In particular, we thank Reviewer Q9VY for highlighting “strong experiments across different modalities that verify their claim”, Reviewer iWNq for highlighting that “improving the communication efficiency of distributed fine-tuning is a promising research direction”, and reviewer xXhU for highlighting “the results show a clear trend, there is little doubt the method works and achieves the claims in the paper”.

---

### Decision · Action_Editors · 2023-01-12

**Recommendation:** Accept as is

**Comment:**

This paper presents a straightforward application of existing methods (e.g. federated learning) to the setting of (distributed) transfer learning. One reviewer complained about lack of novelty, but I think that highlighting that this simple approach can work in the ubiquitous transfer learning setting is valuable. Other reviewers had some initial suggestions primarily around clarity and framing which were addressed. I think this paper makes a meaningful contribution and is worth publishing, since it could potentially open up distributed training to resource-constrained individuals in an incredibly common ML pipeline.

**Audience:**

Yes. Fine-tuning pre-trained models is incredibly common, and most individuals lack supercomputer clusters with many servers with high interconnect. The simple finding in this paper will enable fine-tuning with larger amounts of data parallelism when such high-interconnect GPU servers/clusters are not available. It also contributes to a growing body of work on methods for combining the capabilities of models trained on different data, which is increasingly of interest.

**Claims And Evidence:**

Yes, the method is testing on two realistic models that are commonly fine-tuned in the real world. One reviewer noted that fine-tuning on Common Crawl is uncommon, but the authors expanded their results to include the fine-tuning RoBERTa on GLUE, which is very common and standard. The method is also heavily based on prior work that is tried-and-true.